# Magnetostatic twists in room-temperature skyrmions explored by nitrogen-vacancy center spin texture reconstruction

Y. Dovzhenko[1], F. Casola[1,2], S. Schlotter [3,4], T.X. Zhou [1,3], F. Büttner [4], R.L. Walsworth[1,2], G.S.D. Beach [4] & A. Yacoby[1]

Magnetic skyrmions are two-dimensional non-collinear spin textures characterized by an integer topological number. Room-temperature skyrmions were recently found in magnetic multilayer stacks, where their stability was largely attributed to the interfacial Dzyaloshinskii–Moriya interaction. The strength of this interaction and its role in stabilizing the skyrmions is not yet well understood, and imaging of the full spin structure is needed to address this question. Here, we use a nitrogen-vacancy centre in diamond to measure a map of magnetic fields produced by a skyrmion in a magnetic multilayer under ambient conditions. We compute the manifold of candidate spin structures and select the physically meaningful solution. We find a Néel-type skyrmion whose chirality is not left-handed, contrary to preceding reports. We propose skyrmion tube-like structures whose chirality rotates through the film thickness. We show that NV magnetometry, combined with our analysis method, provides a unique tool to investigate this previously inaccessible phenomenon.

[1] Department of Physics, Harvard University, 17 Oxford Street, Cambridge, MA 02138, USA. [2] Harvard-Smithsonian Center for Astrophysics, 60 Garden Street, Cambridge, MA 02138, USA. [3] John A. Paulson School of Engineering and Applied Sciences, Harvard University, Cambridge, MA 02138, USA. [4] Department of Materials Science and Engineering, Massachusetts Institute of Technology, Cambridge, MA 02139, USA. These authors contributed equally: Y. Dovzhenko, F. Casola. Correspondence and requests for materials should be addressed to A.Y. (email: yacoby@physics.harvard.edu)

Magnetic skyrmions are topological defects originally proposed as being responsible for the suppression of long-range order in the two-dimensional (2D) Heisenberg model[1,2] at finite temperature. The earliest observations of magnetic skyrmions were reported in bulk crystals[3] of noncentrosymmetric ferromagnetic materials at cryogenic temperatures. Recently, a new class of thin film materials has emerged, which support skyrmions at room temperature[4–8]. These results have paved the way towards spintronics applications and call for a quantitative and microscopic characterization of the novel spin textures. However, magnetic imaging of sputtered thin films at room temperature in the presence of variable external magnetic fields represents a serious experimental challenge for established techniques[8].

We address this challenge using a magnetic sensor based on a single nitrogen-vacancy (NV) centre in diamond[9]. We record the projection on the NV axis of the magnetic field produced by the magnetization pattern in the film. This information is sufficient for reconstructing all three components of the magnetic field without the need for vector magnetometry[10]. However, obtaining the underlying spin structure is an under-constrained problem[11]. System-dependent assumptions, e.g. regarding the spatial dependence of a certain spin component[12], may artificially restrict the manifold of solutions compatible with experimental results.

We introduce a method to study such a manifold and show that we can classify all solutions by their helicity. We make use of an energetic argument to require continuity of the structure and discard unphysical solutions. We discover a surprising type of structure that disagrees with previous reports of Dzyaloshinskii–Moriya interaction (DMI)[13,14] in similar materials[15–18].

## Results

**Domain evolution in external magnetic field.** An overview of our scanning magnetometry set-up is shown in Fig. 1a–c (see also Supplementary Note 1 and Supplementary Fig. 1). The sample of interest is deposited on a quartz tip and scanned underneath a stationary diamond pillar, which contains a single NV centre about 30 nm below the surface. An image of a typical diamond pillar of approximately 200 nm diameter is shown in Fig. 1a. The sample consists of a sputtered [Pt (3 nm)/Co (1.1 nm)/Ta (4 nm)] × 10 stack with a seed layer of Ta (3 nm)[6]. We pattern 2-µm diameter discs of this film on the flat surface of a cleaved quartz tip, pictured in Fig. 1c (see Supplementary Fig. 1). All measurements are performed in ambient conditions with a variable bias magnetic field delivered by a permanent magnet and aligned along the NV axis.

In order to identify magnetic features in the patterned discs, we employ a qualitative measurement scheme based on the rate of NV photoluminescence. In the presence of stray magnetic fields perpendicular to the NV axis, fewer red photons are emitted by the NV centre under continuous green excitation[19]. Two photoluminescence scans across the sample at different values of the bias magnetic field are shown in Fig. 1d, e. At 6.5 mT of external magnetic field, we observe a stripe-like modulation of the NV photoluminescence (see Fig. 1d). This pattern is reminiscent of the labyrinth domain arrangement of the local magnetization expected

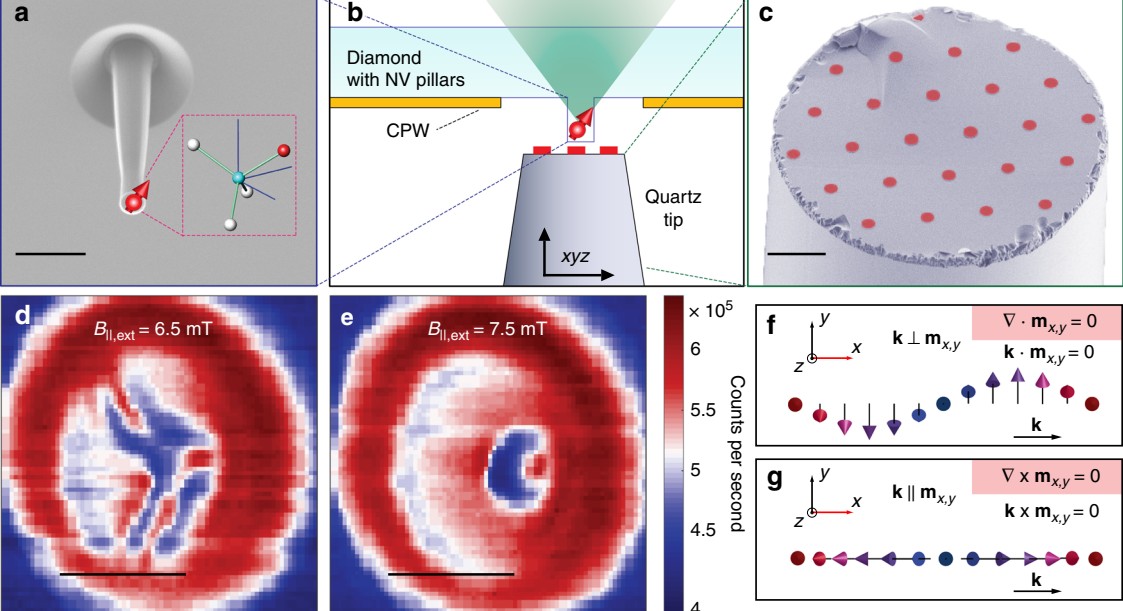

**Fig. 1** Experimental set-up. **a** Electron microscopy image of a typical diamond nanopillar containing a single NV centre approximately 30 nm deep. Rows of such pillars, ~1.5 µm tall, are located inside the gaps of a coplanar waveguide (CPW), which is evaporated on the surface of the diamond (see also **b**). The CPW is used to deliver the microwave excitations necessary to control the NV spin state. The inset shows schematically the geometry of an NV centre in a diamond lattice, pictured in greater detail in Fig. 2d. Scale bar is 1 µm. **b** Sketch of the measurement configuration. A quartz tip with patterned magnetic discs is brought into contact with the diamond nanopillar. The quartz tip and the diamond are mounted on separate stacks of piezo-based positioners and scanners, enabling sub-nanometre movement along all the three *xyz* axes. **c** False-coloured electron microscopic image of a representative quartz tip, where 10 repetitions of a sputtered Pt(3 nm)/Co(1.1 nm)/Ta(4 nm) stack (red) are defined via electron beam lithography and subsequent lift-off as described in Supplementary Note 1. Scale bar is 10 µm. **d, e** NV photoluminescence recorded at 6.5 mT (**d**) and 7.5 mT (**e**) external bias field. The optical excitation power is ~ 100 µW. Higher counts are observed above the magnetic disc due to reflection from the metallic surface. Within the disc boundary, areas with lower counts correspond to large stray magnetic fields perpendicular to the NV axis. Scale bar is 1 µm. **f** Sketch of the Bloch-like spin configuration of a 1D magnetic spiral. Here the local moments of the spiral rotate within a plane that forms an angle $\gamma = \pm\pi/2$ with respect to the propagation vector **k** of the magnetic structure (see text). **g** Structure analogous to **f** for a Néel-like cycloid configuration. Here $\gamma = 0$ ($\pi$) for spins rotating in the anticlockwise (clockwise) direction in the *zx*-plane

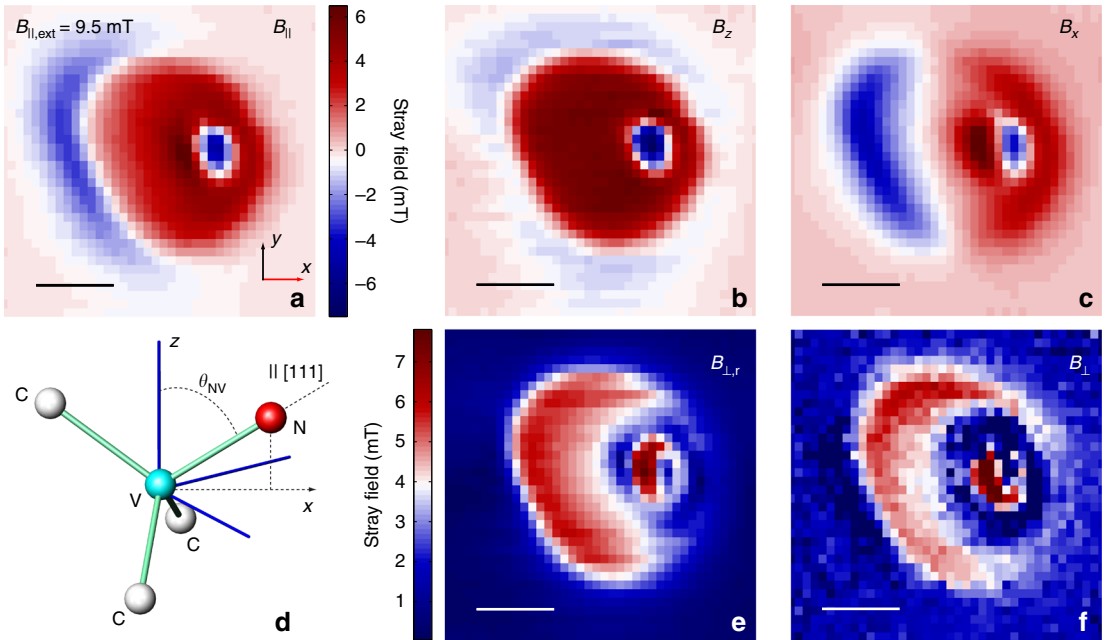

**Fig. 2** Reconstruction of the magnetic stray field components. **a** 2D map of the stray field projection $B_{\parallel}$ on the NV axis (see also **d**). The measurement was performed at a bias field of $B_{\parallel,ext} = 9.5$ mT applied along the [111] diamond axis. **b**, **c** Reconstructed components of the stray field along the z and x-directions, respectively. The z-direction is perpendicular to the magnetic disc. **d** Sketch of the coordination geometry of a nitrogen-vacancy defect in diamond, illustrating the direction parallel to the quantization axis (||) relative to the Cartesian reference frame of the set-up (x, z). Carbon, nitrogen and vacancy sites are labelled C, N and V, respectively. The z axis is orthogonal to the diamond surface. **e**, **f** Reconstructed (**e**) and measured (**f**) magnitude of the stray field perpendicular to the NV centre [111] direction. The measured map is extracted from the spin level mixing of the NV (see Supplementary Note 2). The reconstructed plot is obtained using the procedure outlined in Supplementary Note 2. For all panels, the scale bar is 500 nm

in these materials[6,20]. When the bias field is increased by 1 mT, the labyrinth domains collapse, forming a bubble-like feature shown in Fig. 1e. Our aim in the present paper is to determine the form and nature (see Fig. 1f–g) of the associated spin texture in this high-field regime.

**Vector magnetometry using a single spin sensor**. To extract quantitative information, we use the NV magnetometer to measure 2D spatial maps of the stray field component $B_{\parallel}$ parallel to the NV quantization axis (see Fig. 2a). The measurement plane $\boldsymbol{\rho} = (x, y)$ is parallel to the magnetic film with the NV sensor at a distance $d \sim 30$ nm from this surface. Since no free or displacement currents are present at the NV site, all information about the stray field **B** is contained in the magnetostatic potential $\phi_M$, defined as $\mathbf{B} = -\nabla\phi_M$. It follows that the three spatial components of **B** are linearly dependent in Fourier space, and all components of **B** at a distance $\geq d$ from the film can be obtained numerically from the map at $d$ using upward propagation[11]. These properties of magnetic fields allow us to reconstruct 2D maps for $B_z(\boldsymbol{\rho}, d)$ and $B_x(\boldsymbol{\rho}, d)$ (see Fig. 2b, c) from the 2D scan of $B_{\parallel}(\boldsymbol{\rho}, d)$. In these measurements, the bias field $\mathbf{B}_{ext}$ is aligned with the quantization axis of the NV, which forms an angle $\theta_{NV} \approx 54.7°$ with the axis z normal to the magnetic film surface (see Fig. 2d). We independently confirm the component reconstruction procedure by comparing the reconstructed stray field magnitude perpendicular to the NV axis ($B_{\perp,r}$ in Fig. 2e) to the one extracted from the experiment (see Fig. 2f and Supplementary Note 2 and Supplementary Fig. 2). The good agreement demonstrates our ability to perform vector magnetometry with only one NV orientation.

**Gauge-dependent reconstruction of magnetization**. Because the components of $\mathbf{B}(\boldsymbol{\rho}, d)$ are not independent, they do not contain sufficient information for extracting the underlying spin

structure. We will need additional criteria to narrow down the range of possible solutions. We examine the out-of-plane field $B_z(\boldsymbol{\rho}, d)$, a component that fully preserves all the rotational symmetries of the out-of-plane magnetization. Starting with one magnetic layer and assuming that the local sample magnetization vector $\mathbf{m}(\boldsymbol{\rho}, z) = (\mathbf{m}_{x,y}, m_z)$ is the same throughout the layer thickness $t$, we show (see Supplementary Note 2 and ref. [21]) that $B_z(\boldsymbol{\rho}, d)$ has the following dependence on local magnetization:

$$B_z(\boldsymbol{\rho}, d) = -\frac{\mu_0 M_s}{2}\left(\alpha_z(d, t) * \nabla^2 m_z(\boldsymbol{\rho}) + \alpha_{x,y}(d, t) * \nabla \cdot \mathbf{m}_{x,y}(\boldsymbol{\rho})\right),$$
(1)

where $^*$ denotes convolution in the $x$, $y$-plane, $M_s$ is the maximum value of the saturation magnetization in the disc and we allow $0 \leq \|\mathbf{m}\| \leq 1$ to accommodate spatial dependence of the saturation magnetization of the film. Extension to multilayers is discussed in Supplementary Note 2. The radially symmetric functions $\alpha_z(d, t)$ and $\alpha_{x,y}(d, t)$ are point spread functions, which account for the NV-to-film distance.

Since derivatives commute with convolutions, Eq. (1) is equivalent to Gauss's equation of the form $B_z = -\nabla \cdot \mathbf{F}$, where $B_z$ can be viewed as an effective local charge density and **F** as an effective electric field. The local magnetization components $\mathbf{m}_{x,y}$ and $m_z$ play the role of an effective vector and scalar potential, respectively. In analogy to standard electromagnetism[22], potentials can be uniquely determined by fixing a gauge (see also Supplementary Note 3). Each gauge leads to a different spin helicity[20] $\gamma$ for the magnetic structure **m**. For a simple helical structure, $\gamma$ is the angle between the plane of rotation of the local moments and the propagation vector[23]. For example, spirals have helicity $\gamma = \pm\pi/2$ and are referred to as Bloch configurations in the context of domain walls[12] (see also Fig. 1f). The associated

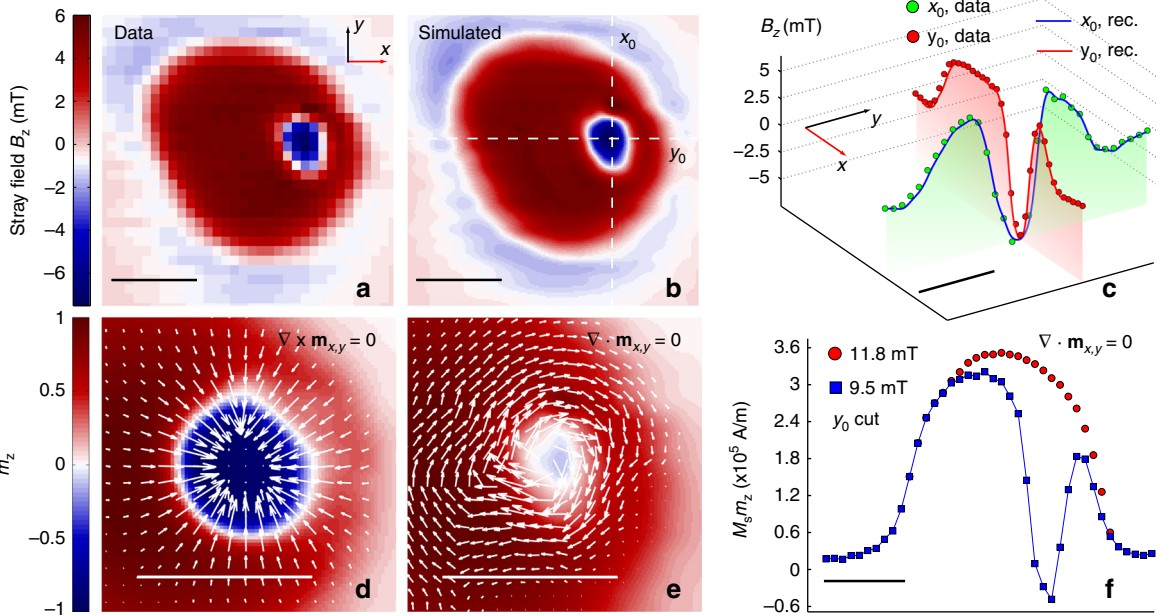

**Fig. 3** Extracting the local magnetic structure of the skyrmion. **a** $z$-component of the stray field from measured data at a bias field of $B_{||,ext} = 9.5$ mT applied along the [111] diamond axis. Since a single component of **B** contains all relevant information, $B_z$ is chosen for comparison with simulations due to its particularly symmetric coupling to $m_z$ (see text). **b** Simulated map of $B_z$ in both the Bloch and the Néel gauge. **c** Cuts along the $x = x_0$ and $y = y_0$ lines shown in **b** (solid lines) and comparison with experimental data in **a** (markers). **d** Magnetic structure obtained in the Néel gauge (see Supplementary Note 3). It preserves normalization of the local magnetization and produces a stray magnetic field that matches the experimental results. The colour map shows the $m_z$ component. White arrows are proportional to the in-plane magnetization. The deviations of the skyrmion profile from a round shape are most likely related to disc edge effects. **e** Plot similar to the one in **d**, obtained by choosing the Bloch gauge. The local magnetization at the centre of the skyrmion in this case is mostly in-plane. **f** Comparison between the reconstructed $M_s m_z$ local magnetization component in the Bloch gauge at two different bias fields (9.5 and 11.8 mT). The $m_z$ profile at saturation (11.8 mT) is used to normalize the local moments for the magnetic structure simulations shown in **d**, **e** (see Supplementary Note 6). From this measurement, we obtain $M_s m_z \simeq 3.6 \cdot 10^5$ A/m at the disc centre (where $m_z = 1$), which agrees with an independently measured value of $M_s m_z = 3.8 \cdot 10^5$ A/m. For all panels, the scale bar is 500 nm, except **c** where it is 400 nm

condition $\mathbf{k} \cdot \mathbf{m} = 0$ for this case can be also expressed as $\nabla \cdot \mathbf{m}_{x,y} = 0$, resembling the Coulomb gauge in electromagnetism[22]. The opposite case is a spin cycloid (see Fig. 1g) with helicity $\gamma = 0$ ($\pi$) representing a Néel-like arrangement of spins[12]. In this case $\nabla \times \mathbf{m}_{x,y} = 0$. We show how to solve Eq. (1) for **m** in both Bloch and Néel gauges in Supplementary Notes 3–5. This gauge approach allows us, for the first time, to systematically identify the complete set of spin structures compatible with local magnetometry data.

For both gauges, we use a numerical variational approach to find a spin structure whose stray field matches the measured field map. The measured field map is shown in Fig. 3a, while a simulated field map from a reconstructed spin structure is plotted in Fig. 3b. We plot cuts through the experimental map and the computed map along $x$ and $y$ axes in Fig. 3c. A 2D plot of the spin structure for the Néel (Bloch) gauge is shown in Fig. 3d (Fig. 3e). In our analysis, we take into account local variations in the saturation magnetization by scaling the magnetization vector **m** to the $m_z$ value obtained in the saturated regime (see Fig. 3f, Supplementary Note 6 and Supplementary Figs. 3 and 4). The two structures in Fig. 3d, e are particular examples chosen from an infinite number of solutions to Eq. (1). These solutions are stable with respect to variation in NV depth, as we demonstrate in Supplementary Note 6 and Supplementary Fig. 5, thus accounting for the inherent uncertainty of NV implantation depth estimation.

A systematic study of the solution manifold requires a way to continuously tune $\gamma$ from the Bloch to the Néel case. To vary the helicity, we start by locally rotating the Bloch solution about the $z$ axis by an angle $\lambda(\phi_N - \phi_B)$, where $\phi_N$ ($\phi_B$) is the local azimuthal angle of the magnetic structure for the Néel (Bloch) configuration.

We then perform a rotation about an axis perpendicular to the resulting local moments so as to preserve its in-plane orientation and at the same time match the measured stray field (see Supplementary Note 7). The parameter $0 \leq \lambda \leq 1$ enables us to move continuously through the manifold. We obtain an ensemble of quantitative, model-independent $m_z(\boldsymbol{\rho}, \lambda)$ profiles for various values of $\lambda$ as shown in Fig. 4a.

**Topology of the solutions**. In order to select the best candidate texture, we study the topology of the 2D vector field $\mathbf{m}(\boldsymbol{\rho}, \lambda)$. For any 2D normalized vector field $\mathbf{n}(\boldsymbol{\rho})$, the topological number $Q$ is defined as:

$$Q = \frac{1}{4\pi} \int dx dy\, \mathbf{n} \cdot \left( \frac{\partial \mathbf{n}}{\partial x} \times \frac{\partial \mathbf{n}}{\partial y} \right). \quad (2)$$

Whenever $\mathbf{n} \parallel z$ at the boundary, any continuous solution $\mathbf{n}(\boldsymbol{\rho})$ must have an integer $Q$ value[24]. Non-integer values of $Q$ occur in the case of a discontinuity, which is energetically costly and unstable[24]. Meanwhile, skyrmions are stable against local perturbations because of the large energetic cost preventing the skyrmion ($Q = 1$) from folding back into the ferromagnetic state ($Q = 0$). We therefore introduce continuity as a criterion for selecting physically allowed solutions. In Fig. 4b, we plot the absolute value of $Q(\lambda)$ for each of the normalized vector fields $\mathbf{n}(\boldsymbol{\rho}, \lambda)$, with $\mathbf{n}$ being the unit vector in the direction of **m**. The number $Q$ can be visualized as the number of times the spin configuration $\mathbf{n}$ wraps around the unit sphere[25]. To illustrate the value of $Q$, in the inset of Fig. 4b we plot the solid angle spanned by $\mathbf{n}$ while moving in the $(x, y)$ plane. We obtain a value for $Q$ approaching $-1$ as $\lambda \to 1$. We therefore identify Néel or nearly-

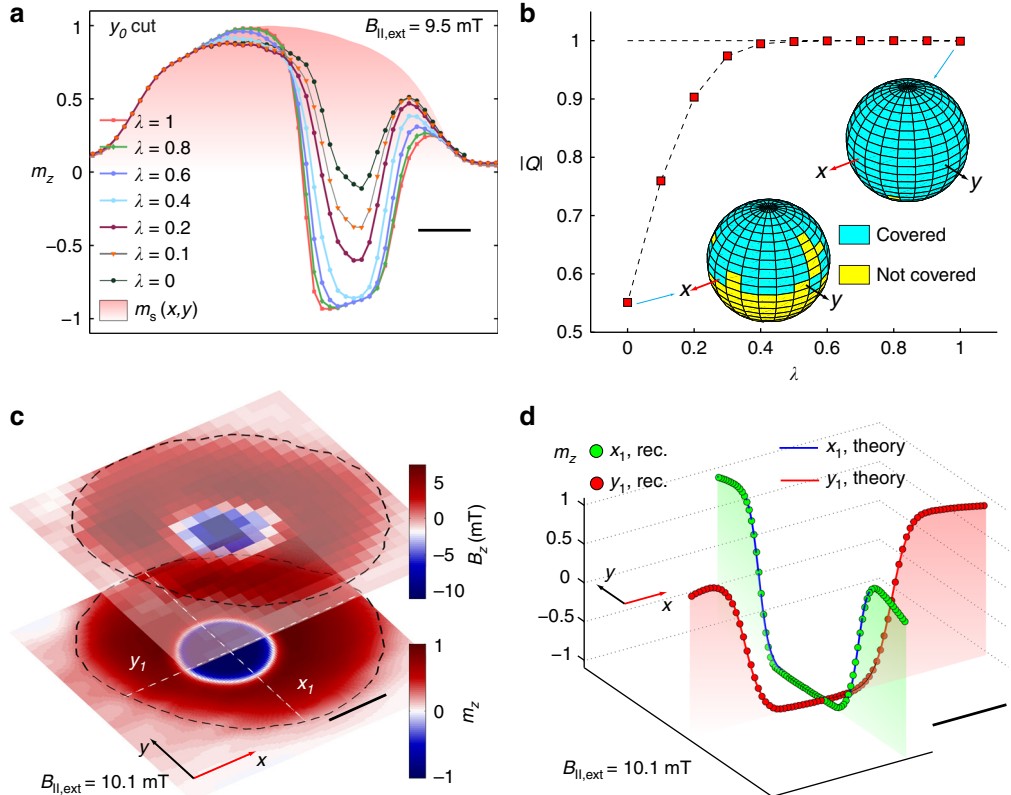

**Fig. 4** Topology of the reconstructed magnetic structure. **a** Continuous tuning of the magnetic structure from the Bloch to the Néel gauge as a function of the parameter $\lambda$ (see text for details). The $m_z$ profiles reported here are cuts along the $y = y_0$ line shown in Fig. 3b. The filled shaded region represents the spatial variation of the normalized saturation magnetization, namely the $m_z$ profile given by the filled red markers in Fig. 3f. **b** Absolute value of the topological number defined in Eq. (2), for each of the spin configurations shown in **a**. The number $Q$ can be visualized as the number of times the vector field wraps around a unit surface. Therefore, the inset shows the stereographic projection of the vector field on a sphere. The image illustrates that only Néel-like configurations have integer $Q$. **c** Map of the $B_z$ component of the stray field (upper sheet) and reconstructed $m_z$ magnetization (lower sheet) for a skyrmion nucleated at the centre of the magnetic disc. The black dashed lines represent the disc boundary. The scan was measured with a bias field parallel to the NV axis of $B_{||,ext} = 10.1$ mT. **d** Comparison of the reconstructed $m_z$ skyrmion profile (markers) with a domain wall model for the skyrmion (solid lines). The profiles are cuts through the $x = x_1$ and $y = y_1$ directions shown in Fig. 4c. Spatial variation of the saturation magnetization is taken into account and the skyrmion profile is observed to be round. The scale bar is 200 nm in **a**, **d**, and 300 nm in **c**.

Néel solutions as the only ones compatible with the measured data.

To make a quantitative comparison of our reconstructed $m_z$ profile in the $\lambda = 1$ case with analytical expressions, we nucleate another skyrmion in the centre of the disc at a bias field of 10.1 mT along the NV axis (see $B_z$ in Fig. 4c). The location of this skyrmion minimizes possible spurious effects caused by the disc edges and allows us to independently test our reconstruction procedure. When comparing line cuts through the $m_z$ profile at the skyrmion centre with existing models proposed in the literature (see Fig. 4d), we observe an out-of-plane magnetization varying in space as $m_z(\tilde{\rho}) = \tanh\left(\frac{\tilde{\rho} - \rho_0}{w/2}\right)$, with $\rho_0$ and $w$ being the skyrmion radius and domain wall width and with $\tilde{\rho}$ being the distance from the skyrmion centre[26]. Our $m_z$ shape are in agreement with the recent first high spatial resolution skyrmion images by X-ray magnetic circular dichroism microscopy and spin-resolved scanning tunnelling microscopy at low temperature[8,26]. The NV-to-film distance $d \sim 30$ nm is too large to extract the domain wall width $w$, but it is sufficient to determine the skyrmion radius $\rho_0 \simeq 210$ nm for the cross sections along the $(x_1, y_1)$ directions shown in Fig. 4d at $B_{||,ext} = 10.1$ mT.

Contrary to expectations, our analysis consistently identifies right-handed ($\gamma = \pi$) Néel-like skyrmions as the only continuous solutions with fixed helicity if we require that the structure does

not vary through the sample thickness. Néel skyrmions are expected from theory when surface inversion symmetry leads to a Rashba-type DMI[27] and the latter dominates over magnetostatic contributions[6]. However, the expected chirality is left-handed ($\gamma = 0$), based on recent X-ray magnetic circular dichroism microscopic measurements of single Pt/Co layers in zero field[8], indirect transport measurements in Pt/Co multilayers through skyrmion movement[6] and studies of domain walls in Pt/Co[15–18], reporting $\gamma = 0$. In contrast with previous data, our skyrmions are not left-handed.

**Variable-gauge solution.** Helicity is dictated by the nature of the energy terms resulting from the breaking of the spatial inversion symmetry along the $z$ axis. In the absence of DMI, Bloch ($\gamma = \pm\pi/2$) configurations are expected[28]. The presence of a chiral DMI term produces $\gamma = 0$ configurations[8]. For thick multilayer dots, even with no DMI the magnetic layers in the vicinity of the top (bottom) surface will experience a breaking of the $z \to -z$ inversion symmetry, favouring Néel spin textures with right-handed (left-handed) chirality[28]. Such twisted structures (also known as Néel caps) reduce the stray field and accordingly the demagnetization energy cost. Néel caps would not be visible with techniques averaging over the sample thickness, such as Lorentz transmission electron microscopy[28,29]. Our technique is most sensitive to the topmost

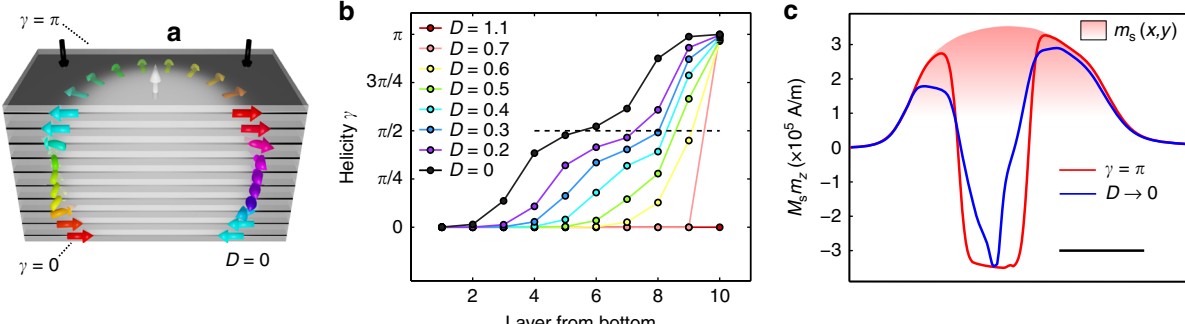

**Fig. 5** Néel caps in magnetic multilayers hosting topological spin structures. **a** Sketch of the magnetic texture obtained via a micromagnetic numerical simulation. The closure domains (i.e. Néel caps[28]) at the top and bottom of the multilayer reduce the demagnetization energy cost with respect to the purely Bloch case. In the simulation $M_s = 10^6$ A/m, $A = 10$ pJ/m, magnetic anisotropy field is 0.2 T and $D_i = 0$ (see Supplementary Note 8). The number of layers and separation is representative of the measured sample. The non-uniformity of $M_s$ and layer thicknesses is not taken into account for this simulation, which may lead to an underestimation of dipolar effects. **b** Local helicity for each one of the ten magnetic layers as the DMI value is varied. The DMI is expressed in mJ/m². Skyrmions with $\gamma \to \pi (\gamma \to 0)$ are present at the top (bottom) of the stack. **c** Cut through of the reconstructed $m_z$ profiles from topologically protected textures that produce a stray field matching the experimental data in Fig. 4c. The red curve corresponds the effective gauge fixed at $\gamma = \pi$ for each layer; the blue curve corresponds to a value of $\gamma = \pi (\gamma = 0)$ for the top (bottom) three layers and $\gamma = \pi/2$ for the four layers in the middle. This red curve approximates the $D_i \to 0$ case depicted in **a**. The filled shaded region represents the spatial variation of the saturation magnetization. The NV depth was again fixed at 30 nm. The scale bar is 500 nm

layers, thus our observation of a right-handed skyrmion is the first to indicate the presence of a Néel cap.

In order to test the energetic stability of skyrmions with changing helicity through the sample thickness, we ran micromagnetic simulations of ten representative proximal magnetic layers, for simplicity with spatially uniform microscopic energy terms (see Fig. 5, details in Supplementary Note 8 and Supplementary Fig. 6). In the limiting case of no DMI ($D_i \to 0$), the top and bottom layers have opposite Néel chiralities, while the intermediate layers are Bloch-like (see Fig. 5a). For small values of the DMI term $D_i$ (see Fig. 5b), right-handed skyrmions are stabilized within the top layers. In order to attempt a comparison of the structure in Fig. 5a with the measured data, we look for a solution with an effective gauge varying through the sample thickness, which is Néel-like for the top and bottom three layers and Bloch- or Coulomb-like for the central part of the multilayer (see Supplementary Note 9 for the details of this procedure). By numerically minimizing the difference between measured and computed field (as done in Supplementary Notes 3 and 4), we obtain the local $m_z$ profile represented by the blue line in Fig. 5c. We compare this solution with the skyrmion solution previously obtained in Fig. 4d (solid red line). The proposed $z$-dependent solution still satisfies $Q \to -1$, but its $m_z$ profile is less sharp. We believe that this shape is due to the variation in skyrmion radius across the multilayer thickness, as suggested by simulations (see e.g. Fig. 5a). The presence of Néel caps and small DMI thus reconciles our data with recent reports of left-handed structures in multilayers and provides evidence in favour of a previously unobserved phenomenon in these films.

## Discussion

Our work is the first example of full vector magnetometry and spin reconstruction performed with a single NV centre. The present method can be applied to any structure with fixed helicity. It also provides an answer to the long-standing magnetometry problem of reconstructing the full set of spin textures from a measured stray field using a general formalism readily applicable to all local magnetometry techniques. The crucial advantage of our technique is its locality and enhanced sensitivity to the topmost magnetic layers. Here we applied these methods to Néel caps in magnetic skyrmions hosted in sputtered Pt (3 nm)/Co

(1.1 nm)/Ta (4 nm) stacks. In contrast with previous work, we rule out purely left-handed Néel solutions in magnetic multilayers. We show that our results are consistent with a previously unobserved twisted structure with vertically evolving chirality and helicity, which is expected from micromagnetic simulations. Our results and methods will be broadly relevant to nanoscale magnetometry and studies of chiral spin textures for room-temperature spintronics applications[6,20,30,31], for example a recently suggested magnetic bobber structure that can coexist with skyrmion tubes[32,33], as well as imaging of current distributions[34,35] in itinerant magnets and magnetic structures in low-dimensional materials[36]. During the review process of this manuscript, we learned of related measurements performed by Legrand and colleagues[37], which have independently confirmed our conclusions regarding the existence of twisted skyrmionic structures in magnetic multilayers.

## Methods

**Sample fabrication**. Magnetic discs are patterned on the flat surface of a cleaved quartz tip, pictured in Fig. 1c, by electron beam lithography (see Supplementary Note 1 and Supplementary Fig. 1). The quartz tip is then mounted on a piezo-electric tuning fork. Monitoring the resonance frequency of the fork allows us to maintain a constant force between the sample and the pillar[9]. We choose the quartz tip diameter to be ~ 50 μm, which allows us to selectively approach an individual NV pillar chosen from a grid of pillars spaced by 50 μm and fabricated on a 2 × 4 mm diamond wafer. We deposit a coplanar waveguide (CPW) on the surface of the diamond, aligned in such a way that rows of pillars reside in gaps. The CPW is used for driving NV centre spin transitions.

**Measurement protocol**. Optical addressing of the NV centre is done through the 50 μm thick diamond. The green laser power used for optical excitation of the NV centre is ~ 100 μW, reduced well below optical saturation in order to avoid heating the sample. A bias magnetic field is delivered by a permanent magnet mounted on a mechanical stage. The magnetic field is aligned parallel to the NV axis, following a procedure based on the NV photoluminescence[21]. This allows us to measure the evolution of magnetic features as function of applied external field, with the field pointing along the NV axis. The nominal value of $M_s$ for the Pt/Co/Ta multilayer film is independently measured using a reference sample placed in the sputtering chamber together with the quartz tip during the deposition process and is found to be $M_s m_z = 3.8 \cdot 10^5$ A/m (see Supplementary Note 6).

**Data availability**. The data that support the findings of this study are available from the corresponding author upon reasonable request.

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

## Acknowledgements

This work is supported by the Gordon and Betty Moore Foundations EPiQS Initiative through Grant GBMF4531. A.Y. and R.L.W. are also partly supported by the QuASAR and the MURI QuISM projects. A.Y. is also partly supported by the Army Research Office under Grant Number W911NF-17-1-0023. The views and conclusions contained in this document are those of the authors and should not be interpreted as representing the official policies, either expressed or implied, of the Army Research Office or the U.S. Government. The U.S. Government is authorized to reproduce and distribute reprints for Government purposes notwithstanding any copyright notation herein. Work at MIT was supported by the U.S. Department of Energy (DOE), Office of Science, Basic Energy Sciences (BES) under Award no. DE-SC0012371 (sample fabrication and magnetic properties characterization). F.C. acknowledges support from the Swiss National Science Foundation (SNSF) grant no. P300P2-158417. S.S. acknowledges the National Science Foundation Graduate Research Fellowship under grant no. DGE1144152. F.B. acknowledges financial support by the German Research Foundation through grant no. BU 3297/1-1. Diamond samples were provided by Element Six (UK). We thank Dr. Marc Warner (Harvard) for helpful ideas in the initial stages of the experiment and Dr. Rainer Stöhr (Harvard–Stuttgart) for technical advice. We thank James Rowland (Ohio State) for fruitful discussions. This work was performed in part at the Center for Nanoscale Systems (CNS), a member of the National Nanotechnology Coordinated Infrastructure Network (NNCI), which is supported by the National Science Foundation under NSF award no. 1541959. CNS is part of Harvard University.

## Author contributions

Y.D., F.C., S.S., G.S.D.B., and A.Y. conceived the experiment. T.Z. and F.C. designed and developed the quartz tips and the diamond. T.Z. optimized the fabrication procedure. S.S. developed the deposition recipes and optimized the magnetic properties of the multi-layers. Y.D. and F.C. performed the experiment. F.C. developed the theoretical model. F.C. and Y.D. performed data analysis. R.L.W., G.S.D.B., and A.Y. provided guidance. F.B. and G.S.D.B proposed the twisted skyrmion model, and F.B. carried out the associated simulations. A.Y. supervised the work. All authors contributed to the writing and the content of the manuscript.
