## [Peer Review File · Nature Communications]

Editorial Note: this manuscript has been previously reviewed at another journal that is not operating a transparent peer review scheme. This document only contains reviewer comments and rebuttal letters for versions considered at Nature Communications

PEER REVIEW FILE

Reviewers' Comments:

Reviewer #2 (Remarks to the Author):

I thank the authors for addressing and giving responses to the comments raised by me and the other referees. Now I understand the versatility of the theoretical framework and how edges and defects would affect the procedure.

As also pointed out by Referee #3 and in Ref.[1], the draft by Dovzhenko et al. does not properly discuss the consequences of the Neel cap on the skyrmion dynamics; thus, I could not clearly understand why the discovery of the Neel cap is so important for the skyrmion physics. However, this issue is well discussed in the preprint from Fert's group [1], in which the authors perform systematic studies and show the impact of the Neel cap on the skyrmion dynamics. After reading Ref.[1], I understood that the Neel cap structure can have a great impact on the skyrmion dynamics in multilayered systems. Although I wanted to see such discussion in this first report of the Neel cap by Dovzhenko et al, I'm now convinced that the experimental discovery of the Neel cap is important and should therefore be appreciated. Hence, I recommend the paper for publication in Nature Communications.

[1] W. Legrand et al., Resolving thickness-dependent reorientation of chiral hybrid textures in magnetic multilayers for spin-torque engineering. arXiv:1712.05978v1 (16 Dec. 2017)

Reviewer #3 (Remarks to the Author):

I support publication of this manuscript in Nature Communications.

Reviewer #4 (Remarks to the Author):

After reading the last round of reviews and rebuttal, it seems that there are still a few sticking

points regarding the analysis and interpretation. I would myself have a few additional questions. However, at this stage I feel that the various relevant communities would greatly benefit to have this paper published without further delay. Therefore, I strongly recommend publication in Nature Communications as is. Here is why:

1) The methodology introduced is new, very interesting and enlightening, and will be of high interest to anyone working in NV magnetometry and more broadly in magnetic microscopy. It answers a number of questions researchers in these fields have had for a long time in terms of reconstruction of M from B, and personally it will be one of the key papers I will recommend to my incoming students from now on.

2) Even though part of the methodology applies only to the skyrmion case as pointed out by previous referees, there are definitely enough details and explanations for the reader to have a good idea of how it could be adapted (or not) in other situations. It is possible that future work will reveal that the technique is not so versatile in situations other than the skyrmion case, but this is such an important topic (knowing what can be inferred from B measurements) that the paper will be extremely useful to many no matter what (see previous point).

3) The results themselves are very exciting and inspiring, in that the measurements clearly indicate a skyrmion of the "wrong" chirality. What is still open to debate is the reason for that apparent anomaly. The authors proposed a model (Neel caps + small DMI) which reconciles their data with previous reports and legitimately claim that this is the first evidence for skyrmions with twisted chirality. I understand the concerns by previous referees that this is not a definitive proof, but because since then the findings has motivated studies by other groups and the proposed structure has been confirmed by at least one group, I think it would be unfair to ask the authors for further control experiments at this stage, and instead I think they deserve credit for making the first case in favor of this twisted structure, which is highly relevant to the skyrmion community.

Responses to Reviewers:

Reviewer #2 (Remarks to the Author):

I thank the authors for addressing and giving responses to the comments raised by me and the other referees. Now I understand the versatility of the theoretical framework and how edges and defects would affect the procedure.

We thank the Referee for the comments and questions, which prompted us to carefully consider our method's limitations.

As also pointed out by Referee #3 and in Ref.[1], the draft by Dovzhenko et al. does not properly discuss the consequences of the Neel cap on the skyrmion dynamics; thus, I could not clearly understand why the discovery of the Neel cap is so important for the skyrmion physics. However, this issue is well discussed in the preprint from Fert's group [1], in which the authors perform systematic studies and show the impact of the Neel cap on the skyrmion dynamics. After reading Ref.[1], I understood that the Neel cap structure can have a great impact on the skyrmion dynamics in multilayered systems. Although I wanted to see such discussion in this first report of the Neel cap by Dovzhenko et al, I'm now convinced that the experimental discovery of the Neel cap is important and should therefore be appreciated. Hence, I recommend the paper for publication in Nature Communications.

We appreciate the Referee's understanding that our work is the first indication of a previously unseen phenomenon, and as such it will require future work to fully understand its implications.

[1] W. Legrand et al., Resolving thickness-dependent reorientation of chiral hybrid textures in magnetic multilayers for spin-torque engineering. arXiv:1712.05978v1 (16 Dec. 2017)

Reviewer #3 (Remarks to the Author):

I support publication of this manuscript in Nature Communications.

We thank the Referee for the time and attention required to review our manuscript, and for the insightful comments about the right-handed nature of our skyrmion, which helped us shape our paper's conclusions.

Reviewer #4 (Remarks to the Author):

After reading the last round of reviews and rebuttal, it seems that there are still a few sticking

points regarding the analysis and interpretation. I would myself have a few additional questions. However, at this stage I feel that the various relevant communities would greatly benefit to have this paper published without further delay. Therefore, I strongly recommend publication in Nature Communications as is. Here is why:

1) The methodology introduced is new, very interesting and enlightening, and will be of high interest to anyone working in NV magnetometry and more broadly in magnetic microscopy. It answers a number of questions researchers in these fields have had for a long time in terms of reconstruction of M from B , and personally it will be one of the key papers I will recommend to my incoming students from now on.

We thank the Referee for appreciating the novelty of our work.

2) Even though part of the methodology applies only to the skyrmion case as pointed out by previous referees, there are definitely enough details and explanations for the reader to have a good idea of how it could be adapted (or not) in other situations. It is possible that future work will reveal that the technique is not so versatile in situations other than the skyrmion case, but this is such an important topic (knowing what can be inferred from B measurements) that the paper will be extremely useful to many no matter what (see previous point).

We agree that extension of our methods will require additional work, and at present the full scope of the method is unknown.

3) The results themselves are very exciting and inspiring, in that the measurements clearly indicate a skyrmion of the "wrong" chirality. What is still open to debate is the reason for that apparent anomaly. The authors proposed a model (Neel caps + small DMI) which reconciles their data with previous reports and legitimately claim that this is the first evidence for skyrmions with twisted chirality. I understand the concerns by previous referees that this is not a definitive proof, but because since then the findings has motivated studies by other groups and the proposed structure has been confirmed by at least one group, I think it would be unfair to ask the authors for further control experiments at this stage, and instead I think they deserve credit for making the first case in favor of this twisted structure, which is highly relevant to the skyrmion community.

We appreciate the Referee's understanding that our work merely indicates, but does not conclusively prove, our suggested structure. As such, we take pride in being able to extract unexpected insights into skyrmion structure from a single scan of magnetic field.